# In Vitro Inhibitory Effect of Silver Diamine Fluoride Combined with Potassium Iodide against Mixed-Species Biofilm Formation on Human Root Dentin

**DOI:** 10.3390/antibiotics13080743

**Published:** 2024-08-07

**Authors:** Jutharat Manuschai, Maki Sotozono, Shoji Takenaka, Niraya Kornsombut, Ryouhei Takahashi, Rui Saito, Ryoko Nagata, Takako Ida, Yuichiro Noiri

**Affiliations:** 1Division of Cariology, Operative Dentistry and Endodontics, Faculty of Dentistry, Graduate School of Medical and Dental Sciences, Niigata University, Niigata 951-8514, Japan; jutharat.m@psu.ac.th (J.M.);; 2Department of Conservative Dentistry, Faculty of Dentistry, Prince of Songkla University, Hat Yai 90112, Songkhla, Thailand

**Keywords:** silver diamine fluoride, potassium iodide, biofilm, root caries

## Abstract

Applying a saturated potassium iodide (KI) solution immediately after silver diamine fluoride (SDF) application may affect the inhibitory effects of SDF on biofilm formation. This study compared the efficacy of 38% SDF with and without KI on preventing mixed-species biofilm formation on human root dentin surfaces and assessed ion incorporation into root dentin. The biofilms, composed of *Streptococcus mutans*, *Lactobacillus rhamnosus*, and *Actinomyces naeslundii*, were grown on specimen surfaces treated with either SDF or SDF + KI. After 24 h, the biofilms were evaluated using scanning electron microscopy, live/dead staining, adenosine triphosphate (ATP) assays, colony-forming unit (CFU) counts, and quantitative polymerase chain reaction. A Mann–Whitney *U* test was used to compare the results between the groups. Ion incorporation was assessed using an electron probe microanalyzer. The relative ATP content in the SDF + KI group was significantly higher than that in the SDF group (*p* < 0.05). However, biofilm morphology and the logarithmic reduction in CFUs and bacterial DNA were comparable across the groups. The SDF + KI treatment resulted in less silver and fluoride ion incorporation than that yielded by SDF alone. The inhibitory effects of SDF and SDF + KI on mixed-species biofilm formation were almost equivalent, although KI application affected the ion incorporation.

## 1. Introduction

Root caries is a remarkable dental problem among older adults, particularly in Japan, where there is an increasing prevalence of this condition [1]. A recent prospective study highlighted that 59.6% of older adults requiring nursing care experience new instances of root caries annually [2]. An important consequence of untreated root caries is tooth loss, which significantly negatively affects oral health-related quality of life. Preventive interventions for root caries are needed for older adults to maintain their dentition and oral function throughout their lives.

Dental caries is a disease that results from the dysbiosis of biofilm adherent to the tooth surface, dominated by acidogenic and aciduric bacteria (e.g., *Streptococcus mutans* and lactobacilli) [3,4]. Research on the microbial etiology of root caries has identified that *Streptococcus* and *Actinomyces* species are predominant in supragingival lesions and that *Actinomyces* species are the only cariogenic bacteria that remain predominant in root caries that progress below the gingival margin [5]. Reducing biofilm formation or reducing specific cariogenic bacteria levels can decrease the likelihood of root caries. Thus, the inhibition of microbial colonization or modulating biofilm quality by pretreatment of root surfaces with potential antibiofilm agents is an approach for root caries prevention [6].

Silver diamine fluoride (SDF) has been widely used in Japanese dental clinics to manage root caries in older adults. Several randomized controlled clinical trials have demonstrated that SDF is effective in arresting root caries [7,8,9]. Moreover, a meta-analysis indicated that annual applications of 38% SDF are more effective than controls in preventing root caries [10]. The efficacy of SDF may be ascribed to its antibiofilm properties, which involve SDF-derived precipitates forming on root dentin surfaces [11]. Silver from SDF integrates into the crystal structure of hydroxyapatite [12], reducing bacterial accumulation [13]. Moreover, root dentin surfaces with an SDF coating inhibit decalcification induced by *S. mutans* by releasing silver and fluoride at the bacteria/dentin interface [14].

Although no clinical trials have reported serious adverse effects related to SDF application, its major drawback is the irreversible black staining from silver precipitation, which poses an esthetic concern for many patients [15,16]. Recently, it has been suggested that using a saturated potassium iodide (KI) solution immediately after SDF can reduce noticeable staining without altering the efficacy of SDF in arresting caries [17,18]. However, reversing staining by reducing the number of silver ions remaining on the tooth surface may impact SDF’s antibiofilm effect. While some studies have revealed that the SDF/KI combination does not reduce antibacterial efficacy compared to SDF alone [19], its effectiveness in inhibiting biofilm formation on the root dentin surface remains uncertain. This uncertainty arises because a previous study focused on single-species biofilms of *S. mutans* [11]. To date, no studies on the SDF combined with KI (SDF + KI) application have tested its effects against mixed-species cariogenic biofilm formation after topical application on root dentin.

Therefore, this study aimed to (i) compare the inhibitory effects of 38% SDF and 38% SDF + KI on mixed-species biofilm formation on human root dentin surfaces using a modified Robbins device (MRD) and (ii) assess silver and fluoride incorporation into the root dentin when treated with SDF and SDF combined with KI. An MRD is an in vitro model system that allows reproducible biofilm formation on tooth substrates under controlled flow conditions. The null hypothesis was that there would be no difference in the growth inhibition of mixed-species biofilms on root dentin surfaces between treatments with SDF and SDF + KI.

## 2. Results

### 2.1. Scanning Electron Microscopy (SEM) and Confocal Laser Scanning Microscopy (CLSM) Observations

SEM images revealed a considerably lower number of biofilm clusters on the root dentin surfaces in both the SDF and SDF + KI groups (Figure 1c,d,g,h) than those formed in the untreated control groups (Figure 1a,b,e,f). There was clear evidence of reduced biofilm formation when the dentin surfaces were pretreated with SDF or SDF + KI. However, no remarkable differences were observed between the SEM images of the SDF and SDF + KI groups.

Corroborating the SEM findings, three-dimensional reconstructed images from live/dead staining indicate that the biofilms in both the SDF (Figure 2e–h) and SDF + KI groups (Figure 2m–p) were thinner than those in the control groups (Figure 2a–d,i–l). The ratio of dead to live cells in the SDF and SDF + KI groups was notably higher than in the corresponding controls (Figure 2q); however, differences between the SDF and SDF + KI groups were not readily apparent.

### 2.2. Relative Adenosine Triphosphate (ATP) Content

Figure 3 shows the percentage of relative ATP content in the biofilms formed on root dentin surfaces in the SDF and SDF + KI groups. The relative ATP content in the SDF group (1.83–4.07) was significantly lower than that in the SDF + KI group (3.65–52.00) (*p* < 0.05).

### 2.3. Viable Bacterial Count

The mixed-species biofilm in the control groups constructed with the MRD flow-cell system comprised mainly viable cells of *S. mutans* and *L. rhamnosus*. The number of *A. naeslundii* was lower than that of the other microorganisms (Table 1). Figure 4 shows that the log reduction in viable *S. mutans* (1.60–3.04) and *L. rhamnosus* (2.13–4.38) in the SDF group was marginally greater than that in the SDF + KI group (0.97–2.37 and 2.01–3.55, respectively); however, a statistically significant difference was not observed (*p* = 0.08 and 0.35, respectively). Moreover, there was no significant difference in the log reduction in viable *A. naeslundii* between the SDF (2.00–2.79) and SDF + KI (2.00–2.70) groups (*p* = 0.67).

### 2.4. Bacterial DNA Quantification

The log reduction in bacterial DNA concentration for each group is shown in Figure 5. The log reduction in *L. rhamnosus* in the SDF group (0.40–2.24) was significantly greater than that in the SDF + KI group (−0.76–1.00) (*p* < 0.05), whereas no significant difference was observed for the log reduction in total bacterial 16S rDNA, *S. mutans*, and *A. naeslundii*.

### 2.5. Ion Incorporation

Figure 6 shows the silver and fluoride incorporation into the root dentin in the SDF and SDF + KI groups compared to that of the untreated control. In the SDF group, there was a concentration gradient of silver and fluoride along the interior (Figure 6b,e), with the highest concentration on the surface (black arrowhead). Moreover, there was a noticeably dense layer of silver deposits on the root dentin superficial surface in the SDF group (Figure 6b), which was absent in the SDF + KI group (Figure 6c). The SDF + KI group showed less ion incorporation into dentin than that of the SDF group (Figure 6c,f). The untreated control did not show silver and fluoride incorporation (Figure 6a,d).

## 3. Discussion

Applying a saturated KI solution to minimize black staining caused by silver-derived precipitates on dentin surfaces may affect the antibiofilm effect of SDF. This study found no remarkable difference in the efficacy of inhibitory biofilm formation on root dentin when pretreated with SDF alone versus SDF combined with KI, although KI application did affect the amount of silver and fluoride ion incorporation. Thus, the null hypothesis could not be rejected. For patients concerned with esthetics, using an SDF/KI combination as an antibiofilm agent appears to be a promising approach for preventing root caries.

In this study, there was clear evidence of reduced biofilm formation when the dentin surfaces were pretreated with SDF or SDF + KI. The percentage of reduction in viable bacterial cells in both the SDF and SDF + KI groups compared to the controls ranged from 90% to 99.99% (log reduction of 1–4). The effectiveness of these materials is further supported by findings from SEM (Figure 1) and CLSM (Figure 2) imaging. The antibiofilm effects appear to be due to the incorporation and release of silver and fluoride ions from the root dentin. These ions interact with hydroxyapatite in teeth to form calcium fluoride and silver phosphate [20]. Additionally, the high polarizing power of silver ions promotes strong bonding with amino acids in dentin proteins [21]. Evidence suggests that the precipitation of silver particles on tooth surfaces suppresses bacterial adhesion [12,13]. Moreover, a previous in vitro study using inductively coupled plasma mass spectrometry and fluoride ion-specific electrodes detected silver and fluoride ions within *S. mutans* cells surrounding SDF-coated tooth surfaces [14], indicating that SDF-treated tooth surfaces release these ions at the biofilm/tooth surface interface [14]. Silver ions are bactericidal metal cations that inhibit biofilm formation by inactivating glucosyltransferase enzymes responsible for glucan synthesis [22]. Glucan synthesized from sucrose is essential for mediating the adhesion of bacterial cells to tooth surfaces [23]. Fluoride promotes remineralization and, at high concentrations, impairs the proton-extruding ATPase enzyme related to the carbohydrate metabolism of acidogenic oral bacteria [24].

Although several studies confirm that SDF is effective and safe for patients, the associated tooth staining with SDF creates esthetic concerns, limiting its use in clinical practice [15,16,25]. To address this issue, discoloration by KI immediately after SDF application is recommended [26,27,28,29]. KI reverses tooth staining by preventing the formation of silver phosphate precipitates and creating soluble silver iodide [21]. Previous studies indicate that KI does not modulate the antibacterial efficacy of SDF against plaque biofilms [9,30]. SDF combined with KI shows a stable antibacterial effect against *S. mutans* at the minimum inhibitory concentration of 20% [31]. The use of SDF combined with KI increases cytocompatibility compared to SDF alone [32,33]. However, research on the inhibitory effects of SDF + KI on biofilm formation is limited [11]. A previous study indicated that the SDF + KI treatment was significantly less effective than 38% SDF alone in reducing *S. mutans* biofilm formation [11]. The inhibition of bacterial adhesion and co-aggregation on tooth surfaces may relate to the number of ions incorporated and released, as discussed earlier. In this study, SDF alone tended to present higher efficacy than when combined with KI. As shown in Figure 2, the biofilm on the SDF-treated surface consists mostly of dead cells. The relative ATP content from the biofilm in the SDF group was significantly lower than that in the SDF + KI group (Figure 3), while no notable differences were observed between the viable bacterial counts of the SDF and SDF + KI groups (Table 1). Electron probe microanalyzer (EPMA) analysis showed that SDF supplemented with KI reduced silver and fluoride incorporation into the root dentin compared to that with SDF alone (Figure 6). This reduction may have impacted the antibiofilm properties of the SDF/KI-treated surface. However, this study indicates that the remaining SDF-derived precipitates on root dentin after the 38% SDF + KI application, acting as a reservoir of silver and fluoride ions, appeared sufficient to inhibit mixed-species biofilm formation. Further work needs to validate the long-term efficacy of this antibiofilm agent with the varied concentration of SDF.

Herein, viable cells were assessed after 24 h in a biofilm model constructed using an MRD flow-cell system. The biofilm community was dominated by *S. mutans*, followed by *L. rhamnosus* and *A. naeslundii* (Table 1), aligning with previous findings [34]. The low number of viable *A. naeslundii* may result from shifts in microbial composition related to plaque pH [35]. Previous studies indicate that the microbial composition in the dynamic stability stage of dental biofilms (pH 7.0) is dominated by *Actinomyces* and non-mutans streptococci, which resemble biofilms on clinically sound enamel surfaces [35,36]. However, an acidic environment from frequent sugar intake decreases the viability of *Actinomyces* and non-mutans streptococci, shifting the dominance to more acidogenic/aciduric species like *S. mutans* and lactobacilli. Even when the plaque pH returns to neutral, the recovery of *Actinomyces* and non-mutans streptococci is slow [35,37]. This slow recovery may explain the reduced growth and low viability of *A. naeslundii* in our 24 h mixed-species cariogenic biofilm model. However, the microbial composition and characteristics of the biofilm community may differ with an increased incubation time. Further work with longer observation is needed to perform.

Demineralized dentin leads to more silver uptake than sound dentin due to highly porous tissue and exposed collagen [38]. Although this study used sound human root dentin as a substrate for SDF application, silver precipitates acting as an ion reservoir [39,40] were sufficient to provide an antibiofilm effect. Under physiological conditions, negatively charged proteins in the organic components of teeth coagulate by binding to silver ions [21]. Interestingly, a previous study shows that SDF-coated root dentin retains and releases the greatest amount of silver and fluoride ions compared to coronal enamel. Releasing ions at the biofilm/tooth interfaces inhibits the metabolic activity of *S. mutans* cells [14]. These results support the idea that SDF-derived components are more readily retained on sound root surfaces due to the abundant organic matter in the root dentin. Thus, SDF or SDF + KI could be beneficial as preventive materials on exposed root surfaces, as well as on an arrested carious dentin.

This study compared the effects of 38% SDF with and without KI on mixed-species cariogenic biofilm formation on human root dentin using an MRD flow-cell system, which more closely mimics natural biofilm behavior than mono-species biofilms or static conditions do. Mixed-species biofilm bacteria exhibit distinct growth rates, gene expression patterns, and altered phenotypes reflected in the enhancement of metabolic capacity, stress tolerance, and community-level signaling [41,42,43,44,45]. Previous research indicates that *S. mutans* within mixed-species biofilms increases the expression of specific genes related to glucan synthesis, remodeling, and binding, leading to variations in matrix construction and biofilm maintenance [41]. This study constructed a cariogenic biofilm with dominant bacterial species implicated in the initiation and progression of root caries [5,46,47,48]. A previous study indicates that the flow variable, simulating the continuous flow of saliva or gingival crevicular fluid, is an important factor that may create differences in both the biofilm structure and composition [49]. The MRD is a reproducible flow-cell system for mixed-species cariogenic biofilm formation, useful for evaluating the efficacy of antibiofilm agents before clinical trials [49,50,51]. This system allows the growth of mature biofilms, preventing the overgrowth of planktonic cells and the accumulation of bacterial metabolites. However, no model fully replicates the dynamics of the oral environment, and the limitations of each biofilm model must be considered when interpreting the research findings [51].

Nevertheless, this model provides valuable information for predicting clinical outcomes. While this biofilm model is an in vitro model providing a stable environment that differs from in vivo conditions, the limitations of such an in vitro study must be considered when interpreting the findings, as natural oral biofilms are more complex and dynamic ecosystems. Further research is needed to validate the efficacy of SDF combined with KI in clinical settings. An in situ biofilm model, reflecting intra-oral situations, is considered as an alternative for further validation.

## 4. Materials and Methods

### 4.1. Study Design and Specimen Preparation

This study assessed the effect of 38% SDF with and without KI on cariogenic biofilm grown on root dentin surfaces for 24 h in terms of biofilm morphology, bacterial viability, and bacterial DNA quantity. For ion incorporation into root dentins, silver and fluoride distribution profiles were assessed using EPMA analysis.

Rectangular root dentin slabs (3 × 3 × 2 mm) were prepared from roots of human upper premolar teeth (Niigata University Research Ethics Committee Approval No. 2022-0069) as previously described, with slight modifications [39]. Briefly, a human root was cut along the vertical plane (thickness) using a low-speed diamond saw to create two pieces of dentin specimens from the buccal and palatal sides. Thereafter, pieces of dentin were cut along the second plane (width and length) using tapered diamond burs. The specimen surfaces were further polished using a 2000-grit silicon carbide paper under water irrigation. To remove the organic tissue, specimens were immersed in 2.5% sodium hypochlorite (NaOCl) for 1 min, followed by ultrasonication with 17% ethylenediaminetetraacetic acid for 1 min and re-immersion in 2.5% NaOCl for 1 min. The specimens were mounted on MRD sampling plugs using a silicone ring (10 mm), and the MRD was sterilized using ethylene oxide.

The number of samples for biofilm analysis was defined by considering an effect size of 2, 0.05 level of significance, and 80% power. The total of 28 paired sampling plugs (N = 56) were then randomly assigned to the test materials in either the SDF or SDF + KI groups. One of the paired sampling plugs was allocated to the experimental group, and the other served as a control (n = 14 per group; 5 samples for viable bacterial count and ATP analysis; 5 samples for DNA quantification; 4 samples for morphological observations) (Figure 7). The samples in the SDF group were treated with 38% SDF (Saforide; Bee Brand Medico Dental, Osaka, Japan) for 4 min, followed by washing with distilled water for 30 s prior to the biofilm challenge. The specimens in the SDF + KI group were subjected to a layer of 38% SDF and agitated using a microbrush for 10 s, instantly followed by the application of a saturated KI solution in accordance with the manufacturer’s instructions (Riva star; SDI, Bayswater, Australia) until the creamy white solution became clear. Next, the reaction products were washed off with distilled water for 30 s. The control group did not receive any treatment.

### 4.2. Bacterial Strains and Culture Conditions

In this study, three cariogenic bacterial strains, *S. mutans* (ATCC 25175), *L. rhamnosus* (ATCC 7469), and *A. naeslundii* (ATCC 12104), were used to create biofilms. Cariogenic bacterial strains were grown overnight from frozen stocks in brain–heart infusion (BHI) broth (Difco Laboratories, Sparks, MD, USA) at 37 °C under anaerobic conditions (80% N_2_, 10% H_2_, and 10% CO_2_). Starter cultures were inoculated into fresh media and cultured for 12 or 24 h under the same conditions. Overnight cultures of the three cariogenic strains in BHI were adjusted and diluted to an optimal optical density at 600 nm of approximately 10^7^ colony-forming units (CFUs) per mL prior to inoculation.

### 4.3. Adjusted Saliva Preparation and Saliva Pellicle Formation

Unstimulated saliva was collected from one of the authors. Saliva samples were diluted (1:10) with sterile Ringer’s solution containing 0.05% cysteine (Sigma-Aldrich; St. Louis, MO, USA). Then, the dilute solution was centrifuged at 10,000× *g*, 4 °C for 10 min, and the supernatant was filter-sterilized [52]. The specimen surfaces were covered with adjusted human saliva (15 mL) for 2 h at 37 °C under aerobic conditions.

### 4.4. Biofilm Formation

The MRD method described in previous studies [50,52,53] was used to assess cariogenic biofilm formation. Following saliva pellicle formation, the mixed suspension containing equal amounts of each organism was pumped into the chamber and kept static for 30 min at 37 °C under anaerobic conditions to encourage initial adhesion. After 30 min, the biofilm developed on the specimen in 1/10 strength BHI broth containing 0.05% sucrose for 24 h under continuous flow at a flow rate of 2 mL/min at 37 °C under anaerobic conditions (Figure 7). After the incubation period, the samples were removed from the chamber and gently washed twice with sterile phosphate-buffered saline (PBS; pH = 7.0).

### 4.5. SEM Observation

Two biofilm samples in each group were fixed with 2.5% glutaraldehyde overnight at 4 °C. Subsequently, the samples were washed twice with PBS and dehydrated with an ascending ethanol series (50–100% [*v*/*v*]). The samples were then dried in a desiccator and sputtered with gold palladium [52]. The biofilms were observed using SEM (EPMA-1610; Shimadzu, Kyoto, Japan) at 300× and 1000× magnifications in a beam scan mode at 12 kV.

### 4.6. CLSM Observation

Two biofilm samples in each group were stained with a fluorescent bacterial viability kit (LIVE/DEAD BacLight Bacterial Viability Kit; Thermo Fisher Scientific, Waltham, MA, USA) at room temperature in the dark for 30 min [53]. Biofilm samples were imaged using CLSM (LSM 700; Carl Zeiss, Oberkochen, Germany) with Ar 488 nm and He-Ne 543 nm lasers. The 510–530 nm and ≥610 nm filters were used to detect SYTO 9 and propidium iodide, respectively. Three-dimensional reconstructed images were created using Imaris software version 9.6.0 (Bitplane AG, Zurich, Switzerland), and the dead and live bacterial cells were assessed. Biofilm samples formed on untreated specimen surfaces were used as control.

### 4.7. ATP Bioluminescence Assay

Specimens were transferred to an Eppendorf tube containing 1 mL PBS, ultrasonicated for 5 min, shaken vigorously for 1 min, and ultrasonicated again for 5 min to detach the biofilm from the dentin surfaces. Following biofilm detachment, the ATP level of the collected biofilm suspension was determined using the BacTiter-Glo microbial cell viability assay (Promega, Madison, WI, USA). Luminescence was measured using a GloMaxVR microplate reader (Promega, Madison, WI, USA). The relative ATP content of each test group was calculated using the following formula:(1)$$\%\;\mathrm{Relative}\;\mathrm{ATP}\;\mathrm{content}=\left(\frac{{\mathrm{ATP}\;\mathrm{level}}_{\mathrm{test}\;\mathrm{material}}}{{\mathrm{ATP}\;\mathrm{level}}_{\mathrm{control}}}\right)\times 100.$$

### 4.8. Viable Bacterial Count

The suspension with detached biofilm cells was homogenized, serially diluted tenfold, and plated on selective agar medium. The number of viable bacteria (CFU/mL) was determined after incubation for 48 h at 37 °C under anaerobic conditions. Selective media for *S. mutans*, *L. rhamnosus*, and *A. naeslundii* included Mitis Salivarius agar with bacitracin [54], modified LBS agar [55], and cadmium sulfate–fluoride–acridine trypticase agar [56,57], respectively, with slight modifications. Log reduction in viable cells of each bacterial species was calculated using the following formula:(2)$${\mathrm{Log}\;\mathrm{reduction}}_{\mathrm{viable}\;\mathrm{bacteria}}=\log_{10}\left(\frac{{\mathrm{CFU}\;\mathrm{count}}_{\mathrm{control}}}{{\mathrm{CFU}\;\mathrm{count}}_{\mathrm{test}\;\mathrm{material}}}\right).$$

### 4.9. DNA Isolation and Quantitative Polymerase Chain Reaction (qPCR)

After washing the biofilm samples, DNA was isolated using the DNeasy PowerSoil Pro Kit (Qiagen, Limburg, The Netherlands) according to the manufacturer’s instructions. Total genomic DNA was quantified and stored at −20 °C before further processing. The quantities of total bacterial DNA and three cariogenic strain DNA were evaluated using real-time PCR.

The sequences of primers used were as follows: total bacteria (16S rDNA) (5′-TCCTACGGGAGGCAGCAGT-3′ and 5′-GGACTACCAGGGTATCTAATCCTGTT-3′) [58,59], *S. mutans* (5′-AGCGTTGTCCGGATTTATTG-3′ and 5′-CTACGCATTTCACCGCTACA-3′) [60], *L. rhamnosus* (5′-AGGTGCTTGCATCTTGATTT-3′ and 5′-CGCCATCTTTCAGCCAAGAA-3′) [61], and *A. naeslundii* (5′-CTGCTGCTGACATCGCCGCTCGTA-3′ and 5′-TCCGCTCGCGCCACCTCTCGTTA-3′) [59,62]. Each reaction mixture (final volume 10 µL) contained 0.5 µL template DNA, 5 µL Power SYBR Green Master Mix (Applied Biosystems, Foster City, CA, USA), forward/reverse primers at a final concentration of 500 nM, and 3.5 µL of ultra-pure water. A StepOnePlus Real-Time PCR System (Thermo Fisher Scientific, Foster City, CA, USA) was used to estimate the numbers of biofilm-forming bacteria via a calibration curve method. Calibration curves were prepared using *S. mutans* (ATCC 25175), *L. rhamnosus* (ATCC 7469), and *A. naeslundii* (ATCC 12104) genomic DNA. Log reduction in DNA concentration (copies/mL) was calculated using the following formula:(3)$${\mathrm{Log}\;\mathrm{reduction}}_{\mathrm{DNA}}=\log_{10}\left(\frac{{\mathrm{DNA}\;\mathrm{copy}\;\mathrm{number}}_{\mathrm{control}}}{{\mathrm{DNA}\;\mathrm{copy}\;\mathrm{number}}_{\mathrm{test}\;\mathrm{material}}}\right).$$

### 4.10. EPMA Analysis

Two dentin discs from the SDF and SDF + KI groups were sectioned longitudinally after removing the biofilm and were embedded in a chemically polymerized resin. The cut surface was serially polished with 2400-grit and 4000-grit SiC papers (Marumoto Struers KK, Tokyo, Japan). Element mappings of silver and fluoride ions were performed to evaluate the penetration of silver and fluoride ions into root dentin according to a previous protocol [53]. A root dentin specimen without treatment and incubation served as a negative control.

### 4.11. Statistical Analyses

All statistical analyses were performed using SPSS version 11.0 (SPSS, Chicago, IL, USA). Statistical significance was set at *p* < 0.05. The assumption of normal distribution of the data was tested using the Shapiro–Wilk test. The Mann-Whitney *U* test was used to compare the biofilm cell viability (relative ATP content, log reduction in viable bacteria) and log reduction in DNA concentration between SDF and SDF + KI groups.

## Figures and Tables

**Figure 1 antibiotics-13-00743-f001:**
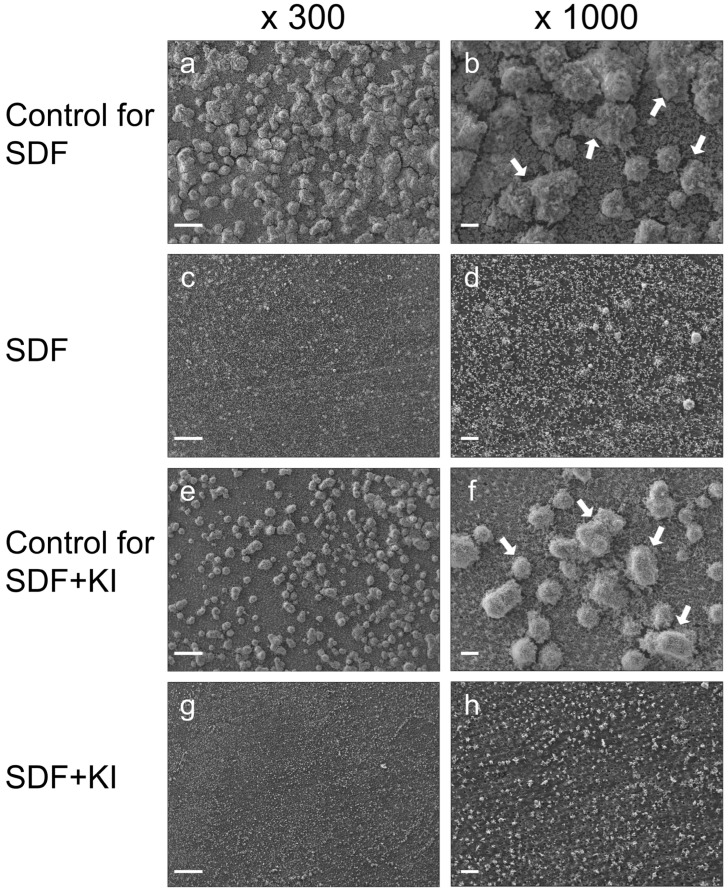
Representative scanning electron microscopy (SEM) images of cariogenic biofilms formed on root dentin surfaces in the SDF (**c**,**d**), SDF + KI (**g**,**h**), and corresponding control groups (**a**,**b**,**e**,**f**) after 24 h of incubation. White arrows indicate the biofilm clusters. SDF: silver diamine fluoride; KI: potassium iodide; scale bars = 100 µm.

**Figure 2 antibiotics-13-00743-f002:**
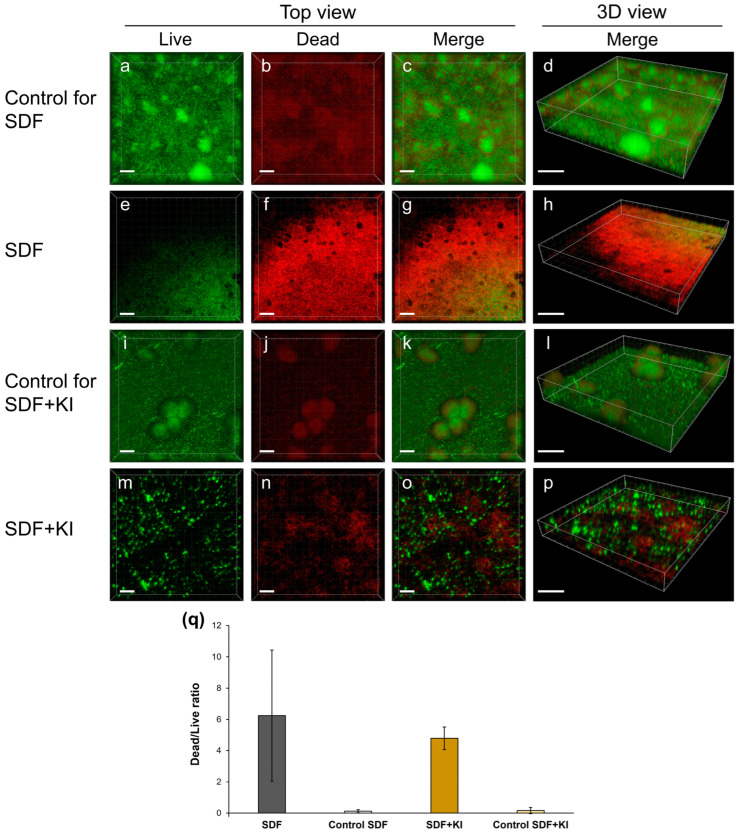
Confocal laser scanning microscopy (CLSM) analysis of mixed-species biofilms consisting of *S. mutans*, *L. rhamnosus*, and *A. naeslundii* formed on root dentin surfaces. (**a**–**p**) Representative three-dimensional reconstructed images corresponding to live/dead staining; scale bars = 20 µm (top view) and 30 µm (3D view). The green signal is due to the SYTO9 dye which indicates live cells, while the red signal is due to propidium iodide which marks the dead cells. (**q**) Ratio of dead to live cells. SDF: silver diamine fluoride; KI: potassium iodide.

**Figure 3 antibiotics-13-00743-f003:**
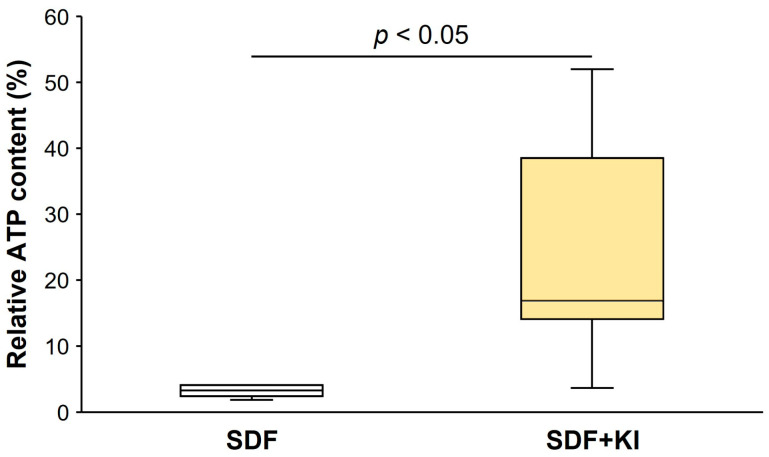
Relative ATP content of biofilm on root dentin surfaces. Data from the control of each test group were used as the standard for calculating the relative content in comparison with the other groups. ATP: adenosine triphosphate; SDF: silver diamine fluoride; KI: potassium iodide.

**Figure 4 antibiotics-13-00743-f004:**
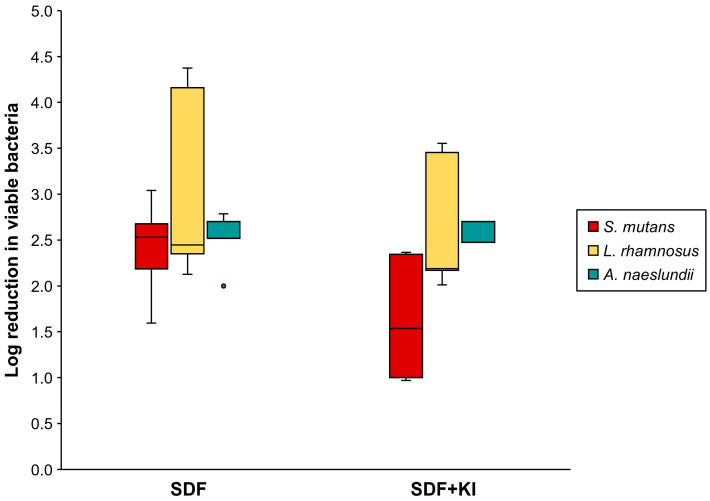
Log reduction in viable cell number (CFU/mL) of *S. mutans*, *L. rhamnosus*, and *A. naeslundii* in the SDF and SDF + KI groups (n = 5). Medians, quartiles, and extreme values are given. SDF: silver diamine fluoride; KI: potassium iodide.

**Figure 5 antibiotics-13-00743-f005:**
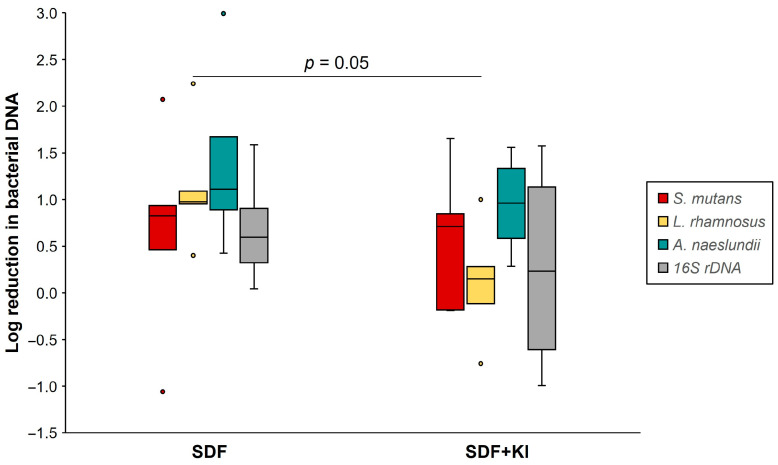
Log reduction in DNA concentration (copies/mL) of *S. mutans*, *L. rhamnosus*, *A. naeslundii*, and 16S rDNA in the SDF and SDF + KI groups (n = 5). Medians, quartiles, and extreme values are given. SDF: silver diamine fluoride; KI: potassium iodide.

**Figure 6 antibiotics-13-00743-f006:**
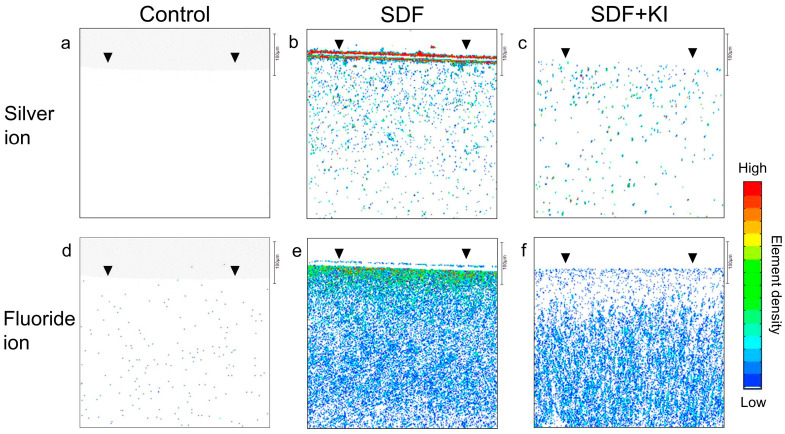
Silver and fluoride distribution profiles in the longitudinal section of root dentin specimens after incubation for 24 h in the control (**a**–**d**), SDF (**b**–**e**), and SDF + KI (**c**–**f**) groups. Arrowheads indicate the disc surface. SDF: silver diamine fluoride; KI: potassium iodide.

**Figure 7 antibiotics-13-00743-f007:**
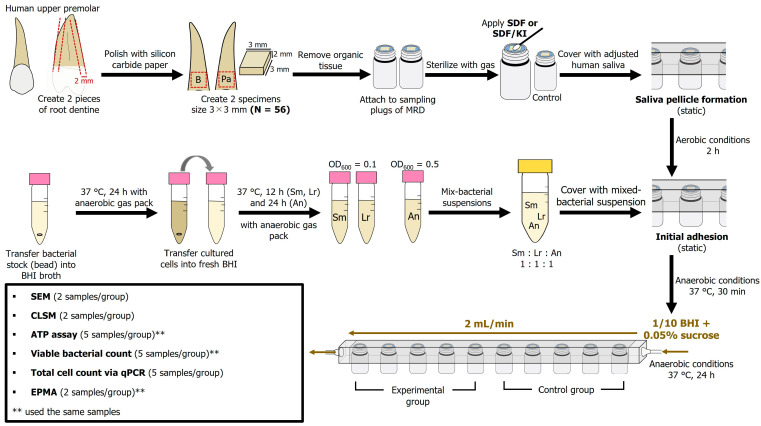
Experimental flow chart. MRD: modified Robbins device; SDF: silver diamine fluoride; KI: potassium iodide; Sm: *S. mutans*; Lr: *L. rhamnosus*; An: *A. naeslundii*; BHI: brain–heart infusion; SEM: scanning electron microscopy; CLSM: confocal laser scanning microscopy; ATP: adenosine triphosphate; qPCR: quantitative polymerase chain reaction; EPMA: electron probe microanalyzer.

**Table 1 antibiotics-13-00743-t001:** Mean, standard deviation (SD), 95% confidence interval (CI), and median of viable bacterial count (log CFU/mL) in each species for the experimental and corresponding control groups (n = 5).

Species	Groups	Viable Bacterial Count (log CFU/mL)				
		Mean	(SD)	95% CI		Median
				Lower Limit	Upper Limit	
*S. mutans*	SDF	4.86	0.39	4.37	5.35	4.62
	Control SDF	7.23	0.10	7.11	7.35	7.21
	SDF + KI	5.16	0.53	4.50	5.81	4.80
	Control SDF + KI	6.75	0.41	6.24	7.25	6.71
*L. rhamnosus*	SDF	4.00	1.07	2.67	5.33	4.51
	Control SDF	6.93	0.25	6.62	7.23	6.95
	SDF + KI	3.69	0.88	2.60	4.78	3.83
	Control SDF + KI	6.31	0.23	6.02	6.59	6.23
*A. naeslundii*	SDF	1.00	0.00	1.00	1.00	1.00
	Control SDF	3.54	0.32	3.15	3.93	3.70
	SDF + KI	1.00	0.00	1.00	1.00	1.00
	Control SDF + KI	3.47	0.29	3.12	3.82	3.48

CFU: colony-forming unit; CI: confidence interval; SD: standard deviation; SDF: silver diamine fluoride; KI: potassium iodide.

## Data Availability

The datasets used and/or analyzed in the current study are available from the corresponding author upon reasonable request.

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
