# Peer review of "In Vitro Inhibitory Effect of Silver Diamine Fluoride Combined with Potassium Iodide against Mixed-Species Biofilm Formation on Human Root Dentin"

_antibiotics, 2024, doi:10.3390/antibiotics13080743_

Round 1

Reviewer 1 Report

Comments and Suggestions for Authors

The article reviewed is of high relevance, scientific, and methodological quality. The results found were relevant and had an important impact on the scientific community. I suggest some adjustments to make it easier to understand, due to the complexity of the information for readers less familiar with the subject of the research. The authors are to be congratulated.

Abstract

Please include statistical methods in the abstract. Report the p-values for the comparisons made. I also suggest creating a graphical abstract to facilitate the understanding of all research stages and groups.

 Introduction

The contextualization is well-written, highlighting the study's relevance, originality, and clinical applicability. The identified knowledge gap in the scientific literature is clear. Ensure that updated studies justify the research. The objective is clearly stated, and a null hypothesis has been added.

 Results

SEM (Scanning Electron Microscopy): Please add labels to the images to align with the information presented in the text. The goal is to provide clarity for readers who are not familiar with the images and their annotations.

Figure 2 is too small to clearly present the information. Please resize it to make the details more visible. If possible, include annotations as requested for the SEM photomicrographs.

For viable bacterial count, include the mean, standard deviation, median, and confidence interval. This will aid other researchers in using your work for sample size calculations.

 Discussion

Ensure that all variables are addressed during the discussion, with a focus on the main variables, as suggested for the study design section to be added in the Materials and Methods section.

 Materials and Methods

The method is well-structured and appropriate. I recommend not omitting or replacing relevant information, but rather providing the related references. There is a gap in following the research stages. Given the extent and quality of the research, I suggest starting this section with a subheading titled "Study Design" to outline the basic formations related to main and secondary variables, factors and variations, formed groups, experimental period, sample size calculation, etc.

Comments on the Quality of English Language

The manuscript has a high quality of English language. There was no difficulty in understanding the texts.

Author Response

Abstract

Please include statistical methods in the abstract. Report the p-values for the comparisons made. I also suggest creating a graphical abstract to facilitate the understanding of all research stages and groups.

The abstract has been included the statistical method and reported the p value. Figure 7 (Experimental flow chart) has been revised to facilitate the understanding of research steps.

Introduction

The contextualization is well-written, highlighting the study's relevance, originality, and clinical applicability. The identified knowledge gap in the scientific literature is clear. Ensure that updated studies justify the research. The objective is clearly stated, and a null hypothesis has been added.

Thank you.

Results

SEM (Scanning Electron Microscopy): Please add labels to the images to align with the information presented in the text. The goal is to provide clarity for readers who are not familiar with the images and their annotations.

We have added labels to the SEM image (Figure 1).

Figure 2 is too small to clearly present the information. Please resize it to make the details more visible. If possible, include annotations as requested for the SEM photomicrographs.

Figure 2 has been resized and included annotations.

For viable bacterial count, include the mean, standard deviation, median, and confidence interval. This will aid other researchers in using your work for sample size calculations.

We have been added the table of viable bacterial count that shows mean, standard deviation, confidence interval, and median values (line 124).

Discussion

Ensure that all variables are addressed during the discussion, with a focus on the main variables, as suggested for the study design section to be added in the Materials and Methods section.

We have added all main variables during the discussion.

Materials and Methods

The method is well-structured and appropriate. I recommend not omitting or replacing relevant information, but rather providing the related references. There is a gap in following the research stages. Given the extent and quality of the research, I suggest starting this section with a subheading titled "Study Design" to outline the basic formations related to main and secondary variables, factors and variations, formed groups, experimental period, sample size calculation, etc.

Thank you for your suggestion. We have added the study design subsection.

Comments on the Quality of English Language

The manuscript has a high quality of English language. There was no difficulty in understanding the texts.

Thank you.

Reviewer 2 Report

Comments and Suggestions for Authors

In this manuscript, “In Vitro Inhibitory Effect of Silver Diamine Fluoride Combined with Potassium Iodide against Mixed-Species Biofilm Formation on Human Root Dentin,” the authors systematically examined whether SDF+KI treatment has a negative impact on the in vitro biofilm inhibitory effect on human root dentin compared to SDF treatment alone. The selected experiments and experimental design are appropriate for the investigation. The methods are detailed and well-documented, allowing for the reproducibility of the experiments. The authors have included the appropriate ethics statement wherever required. The discussion is well-written and points out the limitations of the study. Please address the following major and minor comments:

Major Comments

1) The authors claim that “mixed-species biofilm clusters formed on the root dentin surface.” Figure 1 does not represent the presence of mixed bacterial species. Is there any difference in the shape of the bacterial species that can be shown by further zooming in on the images? If so, please include it as supplementary data.

2) Since it is difficult to fix bacterial cells during confocal imaging, was there any control on when each sample was imaged after staining? If so, please mention this in the methods section.

3) In the current study, the biofilm formation was tested only for 24 hours. In vitro biofilm formation is usually tested after 48 hours. Please explain why 24 hours was selected and include this limitation in the discussion section.

4) The authors have shown that there is more ATP content in the SDF+KI treated dentin compared to SDF-treated dentin. Did the authors see a similar trend when comparing the total viable bacterial count in SDF and SDF+KI treated dentin? Please include those results as supplementary data and discuss them.

5) In lines 200 and 220, the authors mentioned that the precipitates formed are responsible for inhibiting the biofilm formation. Silver and fluoride ions are mainly responsible for the inhibitory effect (citation 35). Thus, the slow release of ions might be the reason for the inhibitory effect. Please include appropriate references showing that precipitates are responsible for the inhibitory effect; otherwise, rewrite to reflect that ions are responsible for the majority of the inhibitory effect.

Minor Comments

1) Please include a better quality image for the SEM image (Figure 1).

2) Throughout the discussion section, please mention the figure numbers showing the actual results while explaining them.

3) In line 288, please correct the typo.

4) In the materials and methods section, 4.1 specimen preparation, the specimen preparation is significantly altered compared to the cited paper. Has any validation been done for the change?

5) In the materials and methods section, 4.5 SEM observation, please include detailed SEM settings.

6) In the materials and methods section, 4.9 DNA Isolation and Quantitative Polymerase Chain Reaction (qPCR), the primer sequence for S. mutans does not match the reference (54). Please explain.

7) The concentration of SDF used in this study was 38%. The reviewer understands that the concentration was selected based on current clinical practice. Are there different concentrations of SDF used in the clinic? As a discussion point, do the authors think increasing the SDF concentration might mitigate the loss of ions with KI application? Please include this in the discussion.

8) In the discussion, please include the next steps that can bridge the gap between in vitro and clinical settings.

Comments on the Quality of English Language

Minor editing for correcting sentence formation is needed.

Author Response

Major Comments

1) The authors claim that “mixed-species biofilm clusters formed on the root dentin surface.” Figure 1 does not represent the presence of mixed bacterial species. Is there any difference in the shape of the bacterial species that can be shown by further zooming in on the images? If so, please include it as supplementary data.

We did not further zoom in to observe each bacterial species within biofilm clusters. Thus, we have revised the sentence in the result section (line 82, 89).   

2) Since it is difficult to fix bacterial cells during confocal imaging, was there any control on when each sample was imaged after staining? If so, please mention this in the methods section.

We used the biofilm samples formed on untreated root dentin surfaces to control the effect of staining procedure on remaining biofilm (line 338-339).

3) In the current study, the biofilm formation was tested only for 24 hours. In vitro biofilm formation is usually tested after 48 hours. Please explain why 24 hours was selected and include this limitation in the discussion section.

We created 24-hour biofilm model to monitor the biofilms at the early stationary phase (reaching maturation stage of biofilm growth). We have mentioned this limitation in the discussion section.

4) The authors have shown that there is more ATP content in the SDF+KI treated dentin compared to SDF-treated dentin. Did the authors see a similar trend when comparing the total viable bacterial count in SDF and SDF+KI treated dentin? Please include those results as supplementary data and discuss them.

The table of viable bacterial count (Table 1) has been added in the result section (line 124). When comparing the total viable bacterial count between SDF and SDF + KI groups, there is no remarkable difference. We have mentioned this result in discussion section (line 197-198). 

5) In lines 200 and 220, the authors mentioned that the precipitates formed are responsible for inhibiting the biofilm formation. Silver and fluoride ions are mainly responsible for the inhibitory effect (citation 35). Thus, the slow release of ions might be the reason for the inhibitory effect. Please include appropriate references showing that precipitates are responsible for the inhibitory effect; otherwise, rewrite to reflect that ions are responsible for the majority of the inhibitory effect.

Thank you for your advice. The sentences have been rewritten (line 202-206; 222-230), and the reference that shows the essential of precipitates for the inhibitory effect has been included.

Minor Comments

1) Please include a better quality image for the SEM image (Figure 1).

We have included a better quality SEM image (Figure 1).

2) Throughout the discussion section, please mention the figure numbers showing the actual results while explaining them.

We have mentioned the figure numbers while explaining them throughout the discussion section.

3) In line 288, please correct the typo.

The typo in line 288 has been corrected.

4) In the materials and methods section, 4.1 specimen preparation, the specimen preparation is significantly altered compared to the cited paper. Has any validation been done for the change?

We performed preliminary study to validate the changed protocol. In organic tissue removal step, we had checked that our protocol totally removes organic tissue without altering the surface properties of specimens.

5) In the materials and methods section, 4.5 SEM observation, please include detailed SEM settings.

The detailed SEM settings have been added (line 328).

6) In the materials and methods section, 4.9 DNA Isolation and Quantitative Polymerase Chain Reaction (qPCR), the primer sequence for S. mutans does not match the reference (54). Please explain.

The reference of the primer sequence for S. mutans has been rechecked and revised (line 365). In addition, the references for all statements have been rechecked throughout the manuscript.

7) The concentration of SDF used in this study was 38%. The reviewer understands that the concentration was selected based on current clinical practice. Are there different concentrations of SDF used in the clinic? As a discussion point, do the authors think increasing the SDF concentration might mitigate the loss of ions with KI application? Please include this in the discussion.

The SDF concentrations available on the clinical practice range from 12% to 38% (80,170 to 254,000 ppm Ag ion concentration). Clinical trials have shown that the efficacy of SDF is concentration-dependent, and proved that the application 38% SDF is safe and more effective than the other concentrations. This study indicates that applying 38% SDF combined with KI is sufficient to provide antibiofilm effects; however, we think changing the SDF concentration may affect the loss of ions with KI application. We have mentioned the further study to validate the efficacy of this antibiofilm agent with varied the concentration of SDF in the discussion section (line 202-206).

8) In the discussion, please include the next steps that can bridge the gap between in vitro and clinical settings.

We have included the next steps to bridge the gap between in vitro and clinical settings in the discussion section (line 258-259).

Comments on the Quality of English Language

Minor editing for correcting sentence formation is needed.

The manuscript has been improved English language.

Reviewer 3 Report

Comments and Suggestions for Authors

The research article titled as "In Vitro Inhibitory Effect of Silver Diamine Fluoride Combined with Potassium Iodide against Mixed-Species Biofilm Formation on Human Root Dentin" is an excellent contribution and I hope it will attract the reader for future research. The authors have for the successfully developed silver diamine fluoride combined with potasium iodide via a simple and cost-effective method utilized. The Silver diamine fluoride combined with potasium iodide were examined using different structural characterization. Moreover, the Silver diamine fluoride combined with potasium iodide demonstrated significant biological activity through in vitro analysis. However, the articles have some minor mistakes that the author needs to correct before the manuscript goes for publication. Some of the suggestions/comments are provided below to further improve the structure of the article. I recommend it for publication (minor revision) after carefully addressing the suggested comments. 

1. Abstract should be restructure with proper order and should be include results as statistical way.

2.   Should be demonstrate MIC, and MBC study of Silver diamine fluoride combined with potasium iodide

3. Should be demonstrate cytotoxicity of Silver diamine fluoride combined with potasium iodide

3.      The author should correct all the typo errors throughout the manuscript.  Eg. hours, minutes, and microbes italics

4.      Author need to improve the English corrections, most of the sentence as improper and hard to read.   

Comments on the Quality of English Language

  Author need to improve the English corrections, most of the sentence as improper and hard to read.   

Author Response

  1. Abstract should be restructure with proper order and should be include results as statistical way.

The abstract has been restructured and reported the p value.

  1. Should be demonstrate MIC, and MBC study of silver diamine fluoride combined with potassium iodide

We have mentioned the minimum inhibitory concentration of silver diamine fluoride combined with potassium iodide in the discussion section (line 187-188).

  1. Should be demonstrate cytotoxicity of silver diamine fluoride combined with potassium iodide

We have mentioned the cytocompatibility of silver diamine fluoride combined with potassium iodide when compared with SDF alone in discussion section (line 188-189).

  1. The author should correct all the typo errors throughout the manuscript. Eg. hours, minutes, and microbes italics

All typo errors throughout the manuscript are corrected.

  1. Author need to improve the English corrections, most of the sentence as improper and hard to read.

The manuscript has been improved the English language.  

Comments on the Quality of English Language

Author needs to improve the English corrections, most of the sentence as improper and hard to read.

The manuscript has been improved the English language.  

Round 2

Reviewer 3 Report

Comments and Suggestions for Authors

I am satisfied all quires with author response. I accept the manuscript for publication